# Fast and Effective Weight Update for Pruned Large Language Models

**Vladimír Boža**                                                                 *boza@fmph.uniba.sk*
*Faculty of Mathematics, Physics and Informatics, Comenius University, Bratislava, Slovakia*

**Reviewed on OpenReview:** *https://openreview.net/forum?id=1hcpXd9Jir*

## Abstract

Pruning large language models (LLMs) is a challenging task due to their enormous size. The primary difficulty is fine-tuning the model after pruning, which is needed to recover the lost performance caused by dropping weights. Recent approaches have either ignored fine-tuning entirely, focusing on efficient pruning criteria, or attempted layer-wise weight updates, preserving the behavior of each layer. However, even layer-wise weight updates can be costly for LLMs, and previous works have resorted to various approximations.

In our paper, we propose a fast and effective weight update algorithm for pruned layers based on the Alternating Direction Method of Multipliers (ADMM). We further extend it with a simple gradual pruning mask selection and achieve state-of-the-art pruning performance across a wide range of LLMs. The code is available at `https://github.com/fmfi-compbio/admm-pruning`.

## 1 Introduction

Large language models (LLMs) (Brown et al., 2020; Zhang et al., 2022; Touvron et al., 2023a;b) have displayed impressive performance in different tasks, but deploying them can be challenging due to their large size and high memory demands. In this work, we introduce a one-shot pruning and weight update algorithm for LLMs that is both fast and effective. Our algorithm produces state-of-the-art results for LLM pruning while imposing minimal computational overhead (Table 1).

Neural networks are usually compressed by either quantization or weight pruning. LLM quantization (Dettmers et al., 2022; Dettmers & Zettlemoyer, 2023; Ahmadian et al., 2023; Xiao et al., 2023) compresses LLMs by storing weights using a small number of bits. On the other hand, pruning compresses models by dropping irrelevant weights (LeCun et al., 1989; Han et al., 2015; Zhu & Gupta, 2018). Pruning can be helpful for LLMs since, during inference, the main bottleneck is memory bandwidth for loading weights to processing unit (Xia et al., 2023). However, the main challenge in deploying LLM pruning is that the network needs to be fine-tuned (Blalock et al., 2020; Liu et al., 2018), which is not feasible with LLMs due to extensive computational and memory footprint. For example, Agarwalla et al. (2024) needed retraining on 45 - 100 billion tokens to recover lost performance by pruning. Also, memory-efficient fine-tuning like LoRA (Hu et al., 2021) is not applicable for LLM weight pruning since we cannot easily merge the low-rank update with the sparsified matrix.

A feasible alternative is one-shot pruning, where one is given a trained model with a small amount of calibration data, and has to compress the model in a single forward pass using limited computational resources. This is typically done via layer-wise pruning, where the pruning problem is split into layer-wise subproblems. In each layer, one aims to select a pruning mask and update weights to minimize reconstruction error. Adaprune (Hubara et al., 2021) solves layer-wise reconstruction by updating weights directly via gradient descent (using Adam optimizer). However, it needs many iterations to achieve convergence. Optimal brain compression (OBC) (Frantar & Alistarh, 2022) removes weights one by one. In each step, it calculates the optimal weight to remove and also the optimal update. However, this approach is also very time-consuming

Table 1: WikiText perplexity of pruned LLaMA-7B. Our ADMM-based methods are superior to previous ones.

| Method | Sparsity | Perplexity |
|---|---|---|
| Dense | 0 % | 5.68 |
| Wanda | 50 % | 7.26 |
| SparseGPT | 50 % | 7.22 |
| ADMM1 | 50 % | 7.20 |
| ADMM-Grad | 50 % | **7.06** |
| Wanda | 60 % | 10.66 |
| SparseGPT | 60 % | 10.51 |
| ADMM1 | 60 % | 9.96 |
| ADMM-Grad | 60 % | **9.22** |
| Wanda | 70 % | 85.77 |
| SparseGPT | 70 % | 26.30 |
| ADMM1 | 70 % | 26.31 |
| ADMM-Grad | 70 % | **18.66** |
| Wanda | 80 % | 5e3 |
| SparseGPT | 80 % | 154.75 |
| ADMM1 | 80 % | 202.04 |
| ADMM-Grad | 80 % | **69.46** |
| Wanda | 2:4 | 11.53 |
| SparseGPT | 2:4 | 11.00 |
| ADMM1 | 2:4 | 10.38 |
| ADMM-Grad | 2:4 | **9.90** |

for pruning LLMs. The first practical approach applicable to LLMs was SparseGPT (Frantar & Alistarh, 2023) using approximations on top of the OBC approach to make the problem feasible, albeit at the cost of decreased reconstruction quality.

Recently, Wanda (Sun et al., 2023) showed that LLMs can be pruned by removing weights with the smallest product of weight magnitude and corresponding input activation norm. This selection approach without the weight update is competitive with SparseGPT on lower sparsities (up to 60%).

**Our results.** In this paper, we introduce an efficient layer-wise weight update algorithm based on alternating direction method of multipliers (ADMM) (Boyd et al., 2011). Our algorithm sidesteps all of the problems of previous solutions. We do not need many gradient descent iterations, nor do we need any heuristic approximation for calculating the weight update. We only need a single inversion of a matrix similar in size to the weight matrix and very few simple iterations to achieve accurate weight updates for a given pruning mask. Furthermore, we extend our algorithm with gradual pruning (Zhu & Gupta, 2018), where in each step, we prune more and more weights. This simple extension allows us to obtain state-of-the-art pruning results at a very low additional cost.

## 2 Preliminaries

### 2.1 Large language models and transformers

Large language models (like Llama) use transformer (Vaswani et al., 2017) architecture and are trained to predict the next word in the text. Transformer consists of multiple repeating blocks. Each block has multihead attention and a feed-forward subblock, which contain multiple linear transformations. Our work focuses on pruning weights in these linear transformations.

## 2.2 One-shot and layer-wise pruning

We consider a scenario of post-training pruning, where we prune an already trained model to a desired sparsity (we assume that the sparsity is the same in each pruned layer).

Since manipulating the whole LLM at once leads to huge computational and memory requirements, we follow the works of Hubara et al. (2021); Frantar & Alistarh (2022; 2023). We prune the LLM during one forward pass (one-shot pruning) and split pruning into multiple layer-wise subproblems. During the forward pass, we capture the calibration inputs for each layer and then prune and update each layer accordingly. More specifically, for each block in the model, we run a forward pass through it, capture inputs for each layer, prune and update weights, and then rerun a forward pass through the whole block to get outputs after pruning.

We are given the original weights $W_\ell$ for each layer and calibration inputs $X_\ell$. Our goal is to find a binary weight mask $M_\ell$ and updated weights $\widehat{W}_\ell$ such that the following reconstruction error is minimized:

$$||X_\ell W_\ell - X_\ell(M_\ell \odot \widehat{W}_\ell)||_2^2$$

For now, we assume that pruning mask $M_\ell$ was found via a separate method and focus only on finding updated weights $\widehat{W}_\ell$.

Assuming that our layer has $n$ output neurons and $m$ inputs, one can just solve $n$ independent linear regressions to solve the problem optimally. Since the mask for each output is different, each one of $n$ outputs requires a separate matrix inversion of the relevant submatrix of $X_\ell^T X_\ell$, which in total takes $O(m^3 n)$ time. This is infeasible even for small neural networks. It is possible to use various approximations to compute updates faster, as done in SparseGPT (Frantar & Alistarh, 2023). However, we demonstrate in our experiments that this compromises the quality of the solution. Another approximation is to not update weights and prune weights with the lowest product of magnitude and input activation norm, as done in Wanda (Sun et al., 2023).

Another possible solution is to update weights iteratively via gradient descent as in Adaprune (Hubara et al., 2021). Here, one update step is proportional to $X_\ell^T X_\ell(M_\ell \odot \widehat{W}_\ell - W_\ell)$. Assuming $X_\ell^T X_\ell$ is precomputed, one update step takes $O(m^2 n)$ time. While this looks much better than solving $n$ linear regressions, Frantar & Alistarh (2023) as well as our own experiments show that one needs many iterations to achieve reasonable convergence.

## 2.3 Alternating Direction Method of Multipliers

Alternating direction method of multipliers (ADMM) (Boyd et al., 2011) is an optimization method for solving problems in the form:

$$\begin{aligned} \text{minimize} \quad & f(x) + g(z) \\ \text{subject to} \quad & Ax + Bz = c \end{aligned}$$

where $f(x)$ and $g(x)$ are convex functions.

ADMM forms the augmented Lagrangian with dual variables $y$ and penalty factor $\rho$:

$$L_\rho(x, z, u) = f(x) + g(z) + y^T(Ax + Bz + c) - \frac{\rho}{2}||Ax + Bz - c||_2^2$$

Typically ADMM is given using scale dual variable $u = y/\rho$ in a form:

$$L_\rho(x, z, u) = f(x) + g(z) + \frac{\rho}{2}(||Ax + Bz - c + u||_2^2 - ||u||_2^2)$$

The Lagrangian is then optimized via the following iterations:

$$x^{k+1} = \arg\min_x f(x) + (\rho/2)||Ax + Bz^k - c + u^k||_2^2$$
$$z^{k+1} = \arg\min_z g(z) + (\rho/2)||Ax^{k+1} + Bz - c + u^k||_2^2$$
$$u^{k+1} = u^k + Ax^{k+1} + Bz^{k+1} - c$$

It can shown that ADMM converges to the optimal solution when $f$ and $g$ are convex and some other mild assumptions hold Boyd et al. (2011).

**Assumption 1.** *The (extended-real-valued) functions $f : \mathbb{R}^n \to \mathbb{R} \cup +\infty$ and $g : \mathbb{R}^m \to \mathbb{R} \cup +\infty$ are closed, proper, and convex.*

**Assumption 2.** *The unaugmented Lagrangian $L_0$ has a saddle point, i.e. there exists $(x^*, z^*, y^*)$ where for all $x, z, y$: $L_0(x^*, z^*, y) \leq L_0(x^*, z^*, y^*) \leq L_0(x, z, y^*)$*

**Theorem 1.** *Let Assumptions 1 and 2 hold. Then:*

- $Ax^k + Bz^k + c \to 0$ *as $k \to \infty$, i.e. iterates approach feasibility.*

- $f(x^k) + g(z^k)$ *approach optimal value as $k \to \infty$*

One application of ADMM is solving constrained optimization over convex set $C$, i.e.:

$$\text{minimize} \quad f(x)$$
$$\text{subject to} \quad x \in C$$

This problem can be rewritten into ADMM form using indicator function $g$, where $g(z) = 0$ if $z \in C$, and $g(z) = \infty$ otherwise:

$$\text{minimize} \quad f(x) + g(z)$$
$$\text{subject to} \quad x - z = 0 \tag{1}$$

In this case, the ADMM update becomes:

$$x^{k+1} = \arg\min_x f(x) + (\rho/2)||x - z^k + u^k||_2^2$$
$$z^{k+1} = \Pi_C(x^{k+1} + u^k) \tag{2}$$
$$u^{k+1} = u^k + x^{k+1} - z^{k+1}$$

Here, $\Pi_C$ is an Euclidean projection operation onto set $C$. Also, note that the $x$ update is just the original unconstrained problem with a simple quadratic penalty term.

## 3 Methods

Here, we propose an alternative solution to finding updated weights in the layer-wise pruning problem. Our solution will have same iteration complexity as gradient descent, but will converge much faster. Recall, that we are given set of calibration inputs $X_\ell$ and mask $M_\ell$ and are looking for $\widehat{W}_\ell$, such that reconstruction error $||X_\ell W_\ell - X_\ell(M_\ell \odot \widehat{W}_\ell)||_2^2$ is minimized.

We observe that when a set of zeroed weights is fixed, valid weight matrices form a convex set $C$. In other words, we are solving the following constrained optimization problem (we omit $\ell$ subscript for clarity):

$$\min_{\widehat{W}} \quad ||XW - X\widehat{W}||_2^2$$
$$\text{subject to} \quad \widehat{W} \odot (1 - M) = 0$$

Our objective is also convex and thus we can use ADMM to solve our optimization problem. We denote our objective as $f(\widehat{W}) = ||XW - X\widehat{W}||_2^2$ and we will use indicator function $g(Z)$ which takes value 0 if $Z \odot (1 - M) = 0$ and $\infty$ otherwise. Using formulation (1) and updates (2), we get following iterations:

$$\widehat{W}^{k+1} = \arg\min_{\widehat{W}} f(\widehat{W}) + (\rho/2)||\widehat{W} - Z^k + U^k||_2^2$$
$$Z^{k+1} = \Pi_C(\widehat{W}^{k+1} + U^k) \tag{3}$$
$$U^{k+1} = U^k + \widehat{W}^{k+1} - Z^{k+1}$$

In our case, $Z$-update is just a projection onto the set of valid matrices, thus:

$$Z^{k+1} = (\widehat{W}^{k+1} + U^k) \odot M \tag{4}$$

Updating $\widehat{W}$ is very similar to ridge regression and can be computed as:

$$\widehat{W}^{k+1} = (X^T X + \rho I)^{-1}(X^T XW + \rho(Z^k - U^k)) \tag{5}$$

**Corollary 1.** *For a fixed calibration input $X$ and mask $M$ iterates 3 (with updates 4, 5) converge to optimal solution for weight update.*

*Proof.* Since our algorithm uses ADMM iterations, we only need to prove that assumptions 1 and 2 hold. $f$ and $g$ are clearly closed, proper, and convex functions; thus, assumption 1 holds.

To show that assumption 2 holds, we need to prove that there exists $(\widehat{W}^*, Z^*, y^*)$ that for all $(\widehat{W}, Z, y)$: $L_0(\widehat{W}^*, Z^*, y) \leq L_0(\widehat{W}^*, Z^*, y^*) \leq L_0(\widehat{W}, Z, y^*)$ where $L_0(\widehat{W}, Z, y) = f(\widehat{W}) + g(Z) + y^T(\widehat{W} - Z)$.

There is a globally optimal solution $\widehat{W}^* = Z^*$ (can be found by independent linear regressions), where: $L_0(\widehat{W}^*, Z^*, y) = f(\widehat{W}) + g(Z)$ and thus $L_0(\widehat{W}^*, Z^*, y) \leq L_0(\widehat{W}^*, Z^*, y^*)$.

If $M_{ij} = 1$ ($\widehat{W}_{ij}$ is unmasked and can have any value), then we set $y_{ij}^* = 0$. If $M_{ij} = 0$, then $Z_{ij}^* = \widehat{W}_{ij}^* = 0$ and we set $y_{ij}^* = -\frac{\partial f}{\partial \widehat{W}_{ij}}(\widehat{W}^*)$.

Then all $\widehat{W}$ and $Z$ derivatives of $L_0$ are zero (or $Z_{ij}^*$ must be 0 due to masking) at $L_0(\widehat{W}^*, Z^*, y^*)$ and since $L_0$ is convex in $\widehat{W}$ and $Z$, then we have a global optimum for given $y^*$ and thus $L_0(\widehat{W}^*, Z^*, y^*) \leq L_0(\widehat{W}, Z, y^*)$. And thus, assumption 2 holds. $\qquad\square$

We can precompute and cache $(X^T X + \rho I)^{-1}$ and $X^T XW$, and then one update iteration has $O(m^2 n)$ complexity, which is the same as the complexity of gradient descent. Note that theoretical results do not say anything about the speed of convergence. In the experimental section, we will show that, in practice, we can get high-quality solutions after very few iterations.

One can also view $\widehat{W}$ update as a way of pulling pruned weights towards zero. Note that for unpruned weights, the penalty term $(\rho/2)||\widehat{W} - (Z^k - U^k)||_2^2$ only limits the step size, but for pruned weights, the value of $Z^k - U^k$ will have different sign than the value of $W$ and thus they will be strongly pulled towards zero.

### 3.1 Mask selection and preconditioning

In the previous section, we described how to update weights when we are given the sparsity mask. Now, we will discuss how to select the mask for pruning.

Wanda (Sun et al., 2023) is a simple rule to select a high-quality mask for pruning LLMs. Instead of selecting weights with the largest value (magnitude pruning), they select weights with the highest product of weight absolute value and input neuron norm, i.e. $|W_{ij}| \cdot ||X_j||_2$. In our implementation, we follow this selection

rule, but we use the norm of inputs as preconditioning. We multiply the weight matrix by feature norms and divide calibration inputs by their feature norms, run the ADMM algorithm, and then normalize the weight matrix back. Note that after the preconditioning, selecting the mask by weight magnitude is equivalent to the Wanda algorithm and that the diagonal of $X^T X$ contains only ones.

Wanda paper also suggests keeping a constant number of weights per output. We found that in our case with weight update, this constraint is actually slightly detrimental, and in our work, we select the top weights for the whole layer.

### 3.2 Gradual pruning

Until now, we considered a scenario where one selects the pruning mask first and then updates weights. Here, we propose a simple extension to our algorithm, which progressively prunes more and more weights and simultaneously computes the weight update. Note, that this still happens during one forward pass, we will just apply multiple iterations to one layer-wise problem.

We adopt cubic sparsity schedule from (Zhu & Gupta, 2018), where sparsity at step $t$ is computed as $s_t = s_f \left(\frac{t}{k_s}\right)^3$, where $s_f$ is final sparsity and $k_s$ is the number of sparsification steps. In each step, we set $s_t$ weights to zero and then proceed with the ADMM update. Note, that only overhead of gradual pruning is just a mask selection added into each step. While $Z^k$ represents the current valid solution, we found that it is slightly better to use $W^{k+1} + U^k$ for selecting weights to prune. This is actually the optimal choice if our constraint (function $g$) was a specific sparsity, not a predefined mask. We summarize our pruning algorithm in Algorithm 1.

We also extend gradual pruning to structured 2:4 sparsity using the following straightforward idea. Our final sparsify will be $s_f = 0.5$. If in step $t$ our target sparsity is $s_t$, then we always keep the two highest elements from each group of four and then prune $2s_t$ weights from the remaining ones.

---

**Algorithm 1** Layerwise gradual pruning with ADMM. Given weight matrix $W$, calibration input $X$, desired sparsity $s_f$, number of iterations $k$, number of sparsification steps $k_s$, dampening factor $\lambda$ (usually 0.1) and penalty factor $\rho$ (usually 1), we prune matrix $W$ to desired sparsity and accurately update weights for the given weight mask.

$norm \leftarrow ||X||_2 + \epsilon$
$W \leftarrow W * norm$
$X \leftarrow X/norm$
$XX \leftarrow X^T X + \lambda I$      // $\lambda$ is usually 0.1
$XXW \leftarrow XX \cdot W$      // precomputation
$XX^{-1} = (XX + \rho I)^{-1}$
**for** $step = 1..k$ **do**
  **if** $step <= k_s$ **then**
    $s_i = s_f \left(\frac{i}{k_s}\right)^3$
    $M \leftarrow$ `mask lowest` $s_i$ `indices`
         `from` $|W + U|$
  **end if**
  $Z \leftarrow (W + U) * M$
  $U \leftarrow U + (W - Z)$
  $W \leftarrow XX^{-1}(XXW + \rho(Z - U))$
**end for**
$W \leftarrow (W + U) * M/norm$

---

### 3.3 Comparison with SparseGPT and Wanda

Compared to SparseGPT (Frantar & Alistarh, 2023), our algorithm does a more accurate weight update since it does not rely on approximation (we also verify this later in the experimental section). It is difficult

to say which mask selection algorithm is better in theory. We gradually prune the whole weight matrix while SparseGPT does optimal selection on group columns of the weight matrix iteratively. But in our experiments our mask selection leads to better results.

Our algorithm can also be thought of as Wanda (Sun et al., 2023) with added weight updates and gradual pruning.

### 3.4 Note on using ADMM with $L_0$ penalty

It is possible to use ADMM to optimize functions under $L_0$ constraint heuristically. This was previously done by Zhang et al. (2018); Ye et al. (2019); Gui et al. (2019). While some of the papers claim that this approach is "systematic", in reality, using ADMM with $L_0$ constraint is just a heuristic since the constraint is not convex. Moreover, in our preliminary experiments, we found that ADMM with $L_0$ constraint is very sensitive to the choice of $\rho$, and for some choices, it will actually run in cycles and not converge.

## 4 Experiments

**General setup.** We implement our algorithms by extending the Wanda (Sun et al., 2023) codebase, which relies on Pytorch and the Huggingface library. Similarly to Wanda and SparseGPT, we use 128 calibration samples from the C4 training dataset (Raffel et al., 2020). We run pruning on a machine with two Quadro RTX 5000 GPUs (each with 16GB of GPU memory). Since we prune layers sequentially in order, we need only to load one layer to GPU memory at one time. This allows us to prune 70B parameter LLaMA models using a relatively small GPU. Unless stated otherwise, we prune for $k = 20$ iterations, using $k_s = 15$ sparsification steps, and set the dampening factor to $\lambda = 0.1$ and ADMM penalty factor $\rho = 1$.

We compare our methods to Wanda (Sun et al., 2023), which does not do weight update and just prunes weights with the lowest product of magnitude and activation norm, and SparseGPT (Frantar & Alistarh, 2023), which uses multiple approximations to select pruned weight and calculating weight updates. For both methods, we use their public implementation and default hyperparameter settings.

**Models and evaluation.** We test our methods on LLaMA (Touvron et al., 2023a) and LLaMA2 (Touvron et al., 2023b) models. Similarly to previous works (Frantar & Alistarh, 2023; Sun et al., 2023), we measure the performance of pruned models on language modeling and zero-shot tasks. Our main focus is perplexity on held-out WikiText (Merity et al., 2016), considered a goto metric for evaluating language model compression (Frantar & Alistarh, 2023). As an additional verification and testing, we use the same seven tasks as Wanda uses from EleutherAI LM Harness (Gao et al., 2021).

### 4.1 Reconstruction error convergence

As a first experiment, we study the quality of our update algorithm. We use a fixed sparsity mask derived using Wanda with 50% sparsity and observe reconstruction error convergence in one layer. We compare our algorithm to gradient-based approaches using Adam and SGD optimizers with varying learning rates. We also compare it to the SparseGPT update (without mask selection) used in the Wanda paper.

The results for selected layers of LLaMA-7b are presented in Figure 1. Our ADMM-based algorithm is superior to both gradient-based algorithms and SparseGPT as it converges almost instantly after computing the initial $X^T X$ matrix inverse. We also note that ADMM works well with the default setting of $\rho = 1$ and does not require learning rate tuning, which starkly contrasts with SGD and Adam, which have different optimal learning rates in different layers.

### 4.2 Weight update quality comparison

In this experiment, we first prune each layer of LLaMA-7B to 60% or 80% sparsity using Wanda mask selection and then update weights either using gradient-based (via Adam) or ADMM update. We select the pruning mask in a single step, i.e., we do not do any gradual mask selection. We test using 1, 10, 20, 50,

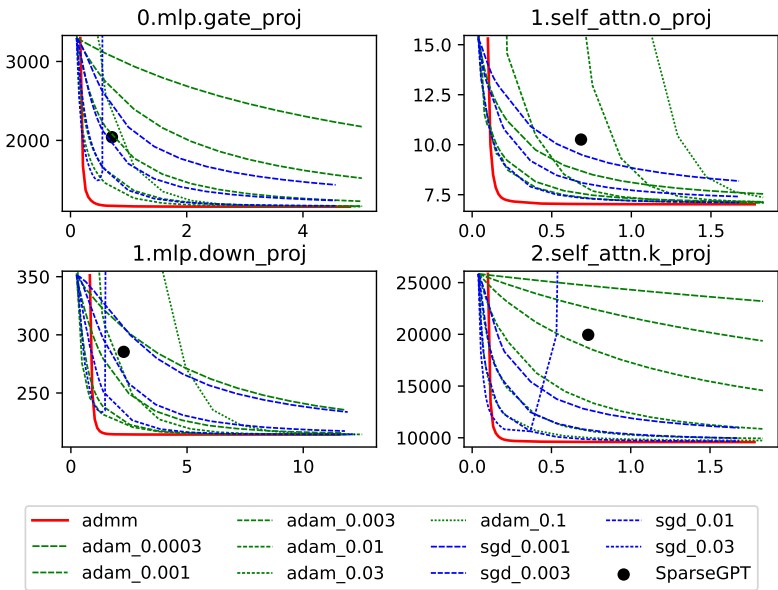

Figure 1: Reconstruction error over time (in seconds) during optimization of weights in selected layers of LLaMA-7B. The mask was derived by Wanda using 50% sparsity. We compare our proposed ADMM algorithm to SGD with momentum and Adam using various learning rates. We also compare to the SparseGPT update. Our ADMM update converges much faster than other methods and is better than the SparseGPT update.

Table 2: Comparision of weight update quality between ADMM and SparseGPT on Llama-7B using 60% sparsity.

| Mask selection | Weight update | Perplexity |
|---|---|---|
| Wanda | SparseGPT | 10.86 |
| Wanda | ADMM | 9.96 |
| SparseGPT | SparseGPT | 10.51 |
| SparseGPT | ADMM | 9.92 |

and 100 update steps. We also test the performance of SparseGPT weight update and, for reference, include results of running SparseGPT with its own gradual mask selection.

We measure perplexity on Wikitext and time overhead (over forward pass) for each update option. Using just one update step, we can almost beat SparseGPT and all gradient-based algorithms (Figure 2). The ADMM update almost converges with ten update steps, while the gradient-based algorithms need more than 100 steps. ADMM is thus clearly a faster and superior weight update algorithm compared to the gradient-based update. Our algorithm also provides a better weight update than SparseGPT weight update, and at 60% sparsity, it is even better than SparseGPT with its own iterative mask selection.

Furthermore, we explicitly compare SparseGPT and ADMM weight updates over different weight masks. We select either Wanda or SparseGPT mask and apply SparseGPT or ADMM weight update (in the case of SparseGPT mask, SparseGPT update is no-op, and for ADMM update, we rewind weights and keep the selected mask). Results are summarized in Table 2. Our ADMM weight update is always better than SparseGPT update. Note that, our mask selection is also better than SparseGPT one (9.22 vs 9.92 perplexity).

### 4.3 Pruning LLaMA-7B

Based on previous observations, we set the number of update iterations to 20, which should provide a pruning overhead similar to SparseGPT (Table 3) and also guarantee reasonable convergence of weight updates. We compare our weight update after mask selection without gradual pruning (ADMM1), our gradual pruning

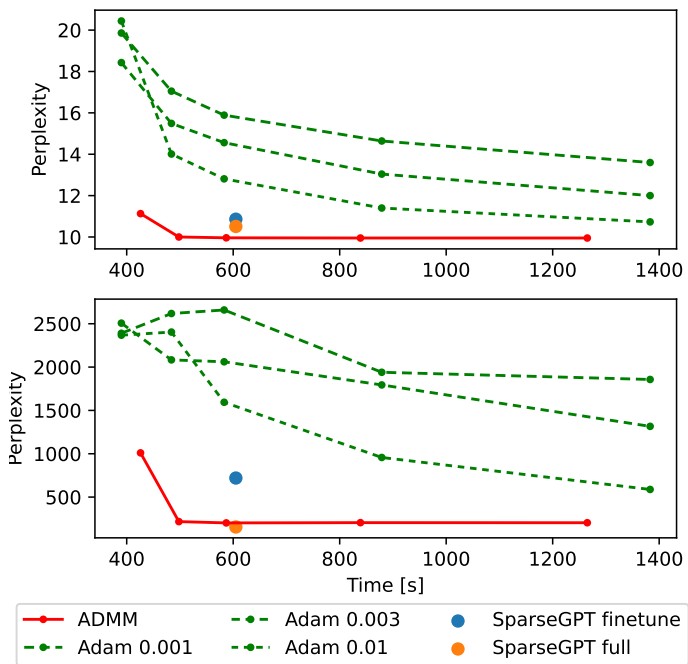

Figure 2: WikiText perplexity vs time overhead for ADMM, Adam, and SparseGPT weight update on LLaMA-7B. We run ADMM and Adam for 1, 10, 20, 50 and 100 update steps and test Adam with various learning rates. The top plot shows 60% sparsity. The bottom one uses 80% sparsity. SparseGPT full refers to normal SparseGPT, which also selects the pruning mask gradually. All other options just update weights over a fixed mask selected by Wanda. Our weight update is better than the one in SparseGPT and better than gradient-based methods.

Table 3: Total pruning time for Llama-7B

| Method | Total seconds |
|---|---|
| Wanda | 245 |
| SparseGPT | 850 |
| ADMM1 | 832 |
| ADMM-Grad | 869 |

algorithm, which computes the mask over 15 iterations (ADMM-Grad) with Wanda and SparseGPT pruning. We prune LLaMA-7b to various sparsities and also with 2:4 structured sparsity. First, we measure Wikitext perplexity (Table 1). We see that our weight update over a fixed Wanda mask (ADMM1) produces better results than any other algorithm at 50%, 60%, and 2:4 sparsities. Note that SparseGPT generates the pruning mask iteratively, which gives it a slight edge in higher sparsities. When selecting the mask gradually, we are superior to all previously developed algorithms, especially at higher sparsities.

Finally, we measure performance on seven zero-shot tasks (we use the same selection as the authors of Wanda): BoolQ (Clark et al., 2019), RTE (Wang et al., 2018), HellaSWAG (Zellers et al., 2019), WinoGrande (Sakaguchi et al., 2021), ARC easy and challenge (Clark et al., 2018), and OpenbookQA (Mihaylov et al., 2018).

Our results (Table 4) show that our algorithm is superior to the previous ones except for the RTE task. We note that results for the RTE task are slightly erratic (e.g. there is better performance at 60% sparsity than at 50%). We attribute this to the small RTE dataset size (277 samples). Notably, we recover 30-40% of the performance drop of SparseGPT on the BoolQ task at 50-70% sparsities and also on WinoGrande task using 50-60% sparsities. When using 2:4 sparsity, we recover 20-25% of the performance drop on WinoGrande and ARC-e tasks.

Table 4: Zero shot accuracies on various tasks during pruning of LLaMA-7B

| Sparsity | Method | BoolQ | RTE | HellaSwag | WinoGrande | ARC-e | ARC-c | OBQA | Mean |
|---|---|---|---|---|---|---|---|---|---|
| 0 % | Dense | 75.05 | 66.43 | 56.92 | 69.93 | 75.34 | 41.89 | 34.40 | 59.99 |
| 50% | Wanda | 71.22 | **55.60** | 51.85 | 66.06 | 69.11 | 36.86 | 28.80 | 54.21 |
| | SparseGPT | 73.05 | 52.34 | 51.21 | 68.42 | 70.70 | 36.43 | 28.60 | 54.39 |
| | ADMM-Grad | **73.63** | 52.34 | **52.33** | **69.13** | **70.74** | **37.88** | **30.20** | **55.18** |
| 60% | Wanda | 69.26 | 59.56 | 43.76 | 62.35 | 62.58 | 30.29 | 25.20 | 50.43 |
| | SparseGPT | 70.7 | **62.09** | 44.84 | 65.58 | 64.14 | 30.97 | 25.20 | 51.93 |
| | ADMM-Grad | **72.41** | 58.84 | **46.61** | **66.77** | **64.52** | **31.65** | **26.20** | **52.43** |
| 70% | Wanda | 59.78 | 58.12 | 28.81 | 50.82 | 32.40 | 18.85 | 14.20 | 37.57 |
| | SparseGPT | 62.35 | **55.95** | 33.77 | 59.35 | 45.70 | 23.97 | 17.20 | 42.61 |
| | ADMM-Grad | **66.05** | 53.79 | **36.29** | **59.74** | **50.84** | **25.50** | **18.60** | **44.40** |
| 80% | Wanda | 37.82 | 48.37 | 26.29 | 48.77 | 27.23 | **20.56** | 13.00 | 31.72 |
| | SparseGPT | 41.89 | **52.70** | 27.83 | 48.38 | 30.30 | 18.77 | **13.40** | 33.32 |
| | ADMM-Grad | **56.14** | **52.70** | **28.75** | **50.74** | **31.56** | 18.94 | 12.40 | **35.89** |
| 2:4 | Wanda | 69.3 | 51.99 | 42.06 | 62.75 | 60.94 | 28.07 | 24.60 | 48.53 |
| | SparseGPT | **70.46** | **60.65** | 42.99 | 64.88 | 61.49 | 30.12 | 23.60 | 50.60 |
| | ADMM-Grad | 70.27 | 55.59 | **44.88** | **66.14** | **64.18** | **30.97** | **25.20** | **51.03** |

Table 5: Perplexity of pruned LLaMA-2 variants on WikiText

| Method | Sparsity | 7B | 13 B | 70B |
|---|---|---|---|---|
| Dense | 0 % | 5.12 | 4.57 | 3.12 |
| Wanda | 50 % | 6.42 | 5.56 | 3.98 |
| SparseGPT | 50 % | 6.51 | 5.63 | 3.98 |
| ADMM-Grad | 50 % | **6.33** | **5.52** | **3.95** |
| Wanda | 60 % | 9.71 | 7.75 | 4.98 |
| SparseGPT | 60 % | 9.58 | 7.80 | 4.98 |
| ADMM-Grad | 60 % | **8.70** | **7.09** | **4.81** |
| Wanda | 80 % | 5e3 | 2e3 | 1e2 |
| SparseGPT | 80 % | 108.87 | 94.23 | 25.86 |
| ADMM-Grad | 80 % | **55.93** | **43.58** | **18.84** |
| Wanda | 2:4 | 11.02 | 8.27 | **5.16** |
| SparseGPT | 2:4 | 10.17 | 8.32 | 5.40 |
| ADMM-Grad | 2:4 | **9.74** | **7.78** | 5.19 |

## 4.4 Pruning LLaMA-2 variants

Our algorithm generalizes and scales to bigger LLMs. We test it on variants of LLaMA-2 at various sparsity levels. Table 5 shows that our method is superior to previous ones, except at 2:4 sparsity on LLaMA2-70B. We note quite a substantial improvement of our algorithm over previous ones at 60% sparsity and also at 2:4 sparsity on 7B and 13B models.

## 5 Related Work

**General neural network pruning.** Post-training network pruning aims to compress neural networks by removing some of their parts (weights, neurons, layers) (LeCun et al., 1989; Han et al., 2015; Blalock et al., 2020; Liu et al., 2018). Pruning criteria vary from simple magnitude pruning (Zhu & Gupta, 2018) to sophisticated second-order approximations (Singh & Alistarh, 2020). Nowadays, there is also a focus on

methods that use limited calibration data and do very little fine-tuning (Frantar & Alistarh, 2022; Hubara et al., 2021).

**LLM pruning algorithms.** Due to sheer LLM size, weight pruning algorithms focused mainly on pruning with limited calibration data and fine-tuning. SparseGPT (Frantar & Alistarh, 2023) solves layer-wise pruning problem using multiple approximations. Wanda (Sun et al., 2023) shows that a simple product of weight magnitude and input activation norm provides competition pruning criterion. DS$\oslash$T (Zhang et al., 2023) provides an iterative mask improvement algorithm.

Another possibility for LLM pruning is structured pruning. One can either remove individual neurons (Ma et al., 2023; Ashkboos et al., 2024), or remove whole layers (Men et al., 2024; Gromov et al., 2024).

**Target specific distillation and tuning.** One can also make neural networks smaller by using knowledge distillation (Hinton et al., 2015). In LLM context, this is usually done with a specific task in mind (Hsieh et al., 2023; Fu et al., 2023; Gu et al., 2023; Ko et al., 2024), where large general model knowledge (logits) is distilled into smaller task-specific model. This is in contrast with our method, which aims to preserve the general ability of the original LLM.

## 6 Conclusions and Future Work

In this work, we presented a simple, fast, and effective post-pruning weight update algorithm based on the alternating direction method of multipliers. We showed that our algorithm converges much faster than any previously available option. Our weight update method is also theoretically sound and does not rely on any heuristical decisions or approximations.

We further improved the pruning performance by doing gradual mask selection and weight updates. This achieves state-of-the-art performance in the layer-wise pruning setting, much better than previous solutions like Wanda or SparseGPT.

Our main limitation is that our update rule runs over dense matrices, and thus, during update computation, we have no time or space savings from potential sparsity. We hope to address this in future work.

Another limitation is that one-shot pruned large models are still inferior to smaller dense ones. The pruning results can certainly be improved by using nonuniform sparsity across layers (Yin et al., 2023); for now, we leave this as a future work. Another option for improvement is to use a more accurate mask selection rule, such as one in Optimal brain surgeon (Hassibi et al., 1993).

Finally, our algorithm provides an efficient update rule for sparse matrices and can be used in some advanced optimizers like FOOF (Benzing, 2022).

### Acknowledgments

This research was supported by grant 1/0538/22 from Slovak research grant agency VEGA.

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
