# OpenReview forum: "Fast and Effective Weight Update for Pruned Large Language Models"
_TMLR — Accepted by TMLR_

### Review · Reviewer_FKAh · 2024-05-14

**Summary Of Contributions:**

The paper provides an algorithm to perform weight updates post-pruning of a
large language model, in order to maintain performance while reducing the
effective model size. This method does not require fine-tuning or
continual training, and is evaluated on the Llama model at three different
sizes (7B, 13B, 70B) and compared to other methods. The
results show that the method is effective at maintaining performance while
reducing the model size, and outperforms other methods in terms of
performance on evaluation datasets.

**Audience:**

Yes

**Claims And Evidence:**

No

**Requested Changes:**

- Please either reduce the claim that fine-tuning an LLM after pruning is not feasible or provide empirical evidence to support this claim.
  - There are several other reasons for not fine-tuning after pruning, for instance, catastrophic forgetting after
    fine-tuning [4]. It would be beneficial to discuss these reasons in the paper.
- Please refrain from using the term "optimal" to describe the method unless there is evidence to support this claim.
- Some of the results suggest that this method (Table 5), and other post-pruning weight update methods,
  lead to inferior performance compared to utilizing a smaller unpruned model.
  It would be beneficial to discuss this in the paper, and why pruning is still a useful technique.
- While the supplemental material provides all the details needed to reproduce the results, it would be beneficial to include
  some of the details in the main body of the paper, such as the hyperparameters used for inference
  in the experiments.

### References

- [4] "Understanding Catastrophic Forgetting in Language Models via Implicit Inference": https://arxiv.org/abs/2309.10105

### Nitpicks

- "showed that LLMs can pruned by removing weights" -> "showed that LLMs can be pruned by removing weights"
- "We prune LLM during one forward pass" -> "We prune the LLM during one forward pass"
- "state-of-the-art performance in layer-wise pruning setting" -> "state-of-the-art performance in the layer-wise pruning setting"

**Strengths And Weaknesses:**

## Strengths

- The evaluations provided of the method are comprehensive, done on several
  model sizes (7B, 13B, 70B), and show that the method is effective at
  performing weight updates after pruning LLMs, while maintaining decent performance.
- The method outperforms similar methods in terms of performance (as measured by perplexity)
  on evaluation datasets, and seems to generalize well as model size increases.
- The method is intuitive and well described, including an algorithm in
  pseudo-code presented in the main body.

## Weaknesses

- The paper claims that fine-tuning an LLM after pruning is not feasible due to the high computational cost.
  However, the paper does not provide any empirical evidence to support this claim, nor cite any work that has shown this to be the case.
  Contrary to this claim, there is work that shows how to fine-tune LLMs efficiently, e.g., [1, 2, 3].
- The work continues to use the term "optimal" to describe their post-pruning method.
  There is no evidence provided that the method is optimal. Optimal in this setting may mean that it is the best possible method.
- The paper could be better structured and be more concise.
  For instance, providing the tables after the conclusion makes it harder to follow the results.
  Additionally, I found the conclusion section to be a bit verbose and could be shortened.
  The paper also does not provide a related works section, however they do
  dive into related works in the introduction and throughout the paper.
  A related works section would help to summarize the related works in one place.

### References

- [1] "LIMA: Less Is More for Alignment": https://arxiv.org/abs/2305.11206
- [2] "LoRA: Low-Rank Adaptation of Large Language Models": https://arxiv.org/abs/2106.09685
- [3] "LoRAPrune: Pruning Meets Low-Rank Parameter-Efficient Fine-Tuning": https://arxiv.org/abs/2305.18403

---

> ### Author Response · Authors · 2024-06-10
> **Authors' response**
>
> Thank you for your review. We are glad that you appreciate the high performance of our method. The following clarifications should address your concerns.
>
> > The paper claims that fine-tuning an LLM after pruning is not feasible due to the high computational cost. However, the paper does not provide any empirical evidence to support this claim, nor cite any work that has shown this to be the case. Contrary to this claim, there is work that shows how to fine-tune LLMs efficiently, e.g., [1, 2, 3].
>
> We added a section to the introduction that shows that fine-tuning needs many iterations (referring to the recent work of Agarwalla et al.) and that low-rank fine-tuning is not applicable here since low-rank update cannot be easily merged with a sparse matrix.
>
> > The work continues to use the term "optimal" to describe their post-pruning method. There is no evidence provided that the method is optimal. Optimal in this setting may mean that it is the best possible method.
>
> We are changing the wording "optimal" to "accurate." We also added notes that ADMM for convex functions converges to the optimal solution, and thus, our update will also converge to the optimal solution. On the other hand, our algorithm uses a fixed number of iterations, so it might not converge to the optimum within a fixed time. Moreover, we are using heuristic (and definitely not optimal) mask selection. Thus, to prevent any over-selling, we will stick to the term accurate.
>
> > The paper could be better structured and be more concise. For instance, providing the tables after the conclusion makes it harder to follow the results. Additionally, I found the conclusion section to be a bit verbose and could be shortened. The paper also does not provide a related works section, however they do dive into related works in the introduction and throughout the paper. A related works section would help to summarize the related works in one place.
>
> We improved table positioning and added related work section.
>
> > Some of the results suggest that this method (Table 5), and other post-pruning weight update methods, lead to inferior performance compared to utilizing a smaller unpruned model. It would be beneficial to discuss this in the paper, and why pruning is still a useful technique.
>
> It is true that a smaller dense model is still more accurate than the one-shot pruned large model. However, this was not a primary objective of our paper. Our main intention (as in previous similar works accepted at ICML or ICLR) was to show that layer-wise weight update could be done using a theoretically sound technique without using various heuristics and that it could achieve competitive results.
>
> Moreover, we tested our pruning algorithm with 40% sparsity, and Llama-2-13B achieved 4.99 perplexity, which is lower than dense Llama-2-7B (5.12 perplexity). This shows that our approach is getting close to Pareto optimality and just needs a little push. Nevertheless, we mention this performance gap in the limitations section in the conclusion.
>
> > While the supplemental material provides all the details needed to reproduce the results, it would be beneficial to include some of the details in the main body of the paper, such as the hyperparameters used for inference in the experiments.
>
> We added all algorithm hyperparameters to the experimental section.

---

### Review · Reviewer_qDYv · 2024-06-04

**Summary Of Contributions:**

This paper adopted ADMM for weight pruning of LLM. ADMM has advantages over existing methods both in terms of the speed and accuracy. For example, ADMM has the same complexity as gradient descent-based methods for one step, but it converges much faster in practice. The optimization problem is cast as an ADMM problem and the update equation and algorithm are presented. The experimental results show the superior performance to other existing weight pruning methods in a similar setting.

**Audience:**

Yes

**Broader Impact Concerns:**

No concerns.

**Claims And Evidence:**

Yes

**Requested Changes:**

It would be good to mention if the performance degradation by the pruning method is acceptable for downstream applications.

It would be good to have discussions on the comparison with distillation/SFT. There are methods to distill and finetune LLM and it could potentially provide better model with a smaller size. It is probably the case that the proposed method is best-in-class in the particular setting, but it would also be important to compare it with methods that are possible with a milder condition.

It would be good to have more detailed explanations on baseline methods to further contrast the differences between methods and to make sure that the comparisons are apples-to-apples and it would also be good to have more detailed explanations on experimental conditions including points mentioned in the weaknesses section above.

**Strengths And Weaknesses:**

Strengths

This paper shows how to adopt ADMM for weight pruning of LLM and provide experimental results showing its effectiveness compared to other approaches. It is a theoretically more sound approach compared to other iterative methods such as gradient-based methods. It also practically works very well -- faster and more accurate.

It is tested on several benchmarks and it supports their claim.

A simple extension to ADMM (gradual pruning) improves the naive application of ADMM significantly. It is simple and effective.


Weaknesses

Although it is clearly better than existing methods evaluated in the paper, but the gap to the original dense model seems to be large. It is not clear if the number is good enough and the resulted model is useful for some real applications.

All of the details of experiments are not explained in the paper. For example, how to obtain the calibration input for each weight is not very clear. The architecture of LLaMA is not explained at all. Readers could go the original paper to understand it, but I think it is an important detail to be explained in the paper, too.

---

> ### Author Response · Authors · 2024-06-10
> **Authors' response**
>
> Thank you for your review. We are glad that you find our method sound and see its practical gains. The following clarifications should address your concerns.
>
> > Although it is clearly better than existing methods evaluated in the paper, but the gap to the original dense model seems to be large. It is not clear if the number is good enough and the resulted model is useful for some real applications.
>
> It is true that a smaller dense model is still more accurate than the one-shot pruned large model. However, this was not a primary objective of our paper. Our main intention (as in previous similar works accepted at ICML or ICLR) was to show that layer-wise weight update could be done using a theoretically sound technique without using various heuristics and that it could achieve competitive results.
>
> Moreover, we tested our pruning algorithm with 40% sparsity, and Llama-2-13B achieved 4.99 perplexity, which is lower than dense Llama-2-7B (5.12 perplexity). This shows that our approach is getting close to Pareto optimality and just needs a little push.
> Nevertheless, we mention this performance gap in the limitations section in the conclusion.
>
> > All of the details of experiments are not explained in the paper. For example, how to obtain the calibration input for each weight is not very clear. The architecture of LLaMA is not explained at all. Readers could go the original paper to understand it, but I think it is an important detail to be explained in the paper, too.
>
> We expanded the preliminaries section to explain how calibration inputs are obtained during the forward pass. We state in the experimental section that we use 128 calibration samples from C4 (similar to SparseGPT and C4). We added short description of transformer to preliminaries.
>
> > It would be good to have discussions on the comparison with distillation/SFT. There are methods to distill and finetune LLM and it could potentially provide better model with a smaller size. It is probably the case that the proposed method is best-in-class in the particular setting, but it would also be important to compare it with methods that are possible with a milder condition.
>
> Yes, these techniques can provide a better task-specific model. We added a short discussion about these methods in the related work section.
>
> > It would be good to have more detailed explanations on baseline methods to further contrast the differences between methods and to make sure that the comparisons are apples-to-apples and it would also be good to have more detailed explanations on experimental conditions including points mentioned in the weaknesses section above.
>
> We added a short explanation of SparseGPT and Wanda to both the preliminaries and experimental sections.

---

### Review · Reviewer_tt5c · 2024-06-05

**Summary Of Contributions:**

The authors propose the use of Alternating Direction Method of Multipliers(ADMM) for layerwise pruning of large language models(LLM). They show that their method is more effective than previous state-of-art approaches like sparseGPT and Wanda in pruning, giving models with better text perplexity and zero-shot accuracies on various tasks at different levels of sparsity.

**Audience:**

Yes

**Claims And Evidence:**

Yes

**Requested Changes:**

- Explanation or modification of the term "optimal" in the title of the paper.

**Strengths And Weaknesses:**

- The proposed algorithm works very well, outperforming state-of-art pruning algorithms like Wanda and SparseGPT in producing models with better text perplexity and zero-shot prediction accuracies at different levels of sparsity.

- In terms of speed the proposed method is also fast, on par with the layerwise optimization approach SparseGPT but not as fast as Wanda.

- One potential weakness of the paper is its novelty. The paper largely follows the optimization framework of SparseGPT, replacing the inner optimizer with ADMM. The mask selection heuristic largely follows from the Wanda algorithm. There are relatively few new technical contributions, even as the method is very effective.

- The title of the paper says fast and optimal weight updates, but in what sense are the weight updates "optimal"? I don't see any explanation of this in the paper.

---

> ### Author Response · Authors · 2024-06-10
> **Authors' response**
>
> Thank you for your review. We are glad that you appreciate the high performance of our method. The following clarifications should address your concerns.
>
> > The title of the paper says fast and optimal weight updates, but in what sense are the weight updates "optimal"? I don't see any explanation of this in the paper.
>
> We are changing the wording "optimal" to "accurate." We also added notes that ADMM for convex functions converges to the optimal solution, and thus, our update will also converge to the optimal solution.
> On the other hand, our algorithm uses a fixed number of iterations, so it might not converge to the optimum within a fixed time.
> Moreover, we are using heuristic (and definitely not optimal) mask selection. Thus, to prevent any over-selling, we will stick to the term accurate.

---

> ### Author Response · Authors · 2024-06-17
> **One more note about new contributions.**
>
> We believe that replacing SparseGPT with ADMM is actually a new contribution.
> The core problem is efficiently solving multiple linear regressions over the same superset of features.
> SparseGPT opts for heuristics since no efficient solution was known at the time.
> We provide an efficient and accurate solution for this problem. It is also possible that solving "multiple linear regressions over the same superset of features" is an interesting problem outside of the neural network pruning area.

---

### Author Response · Authors · 2024-06-10
**Revision**

We thank the reviewers for their constructive, insightful, and detailed feedback.

Based on comments from two reviewers, we are changing the wording in the title from "optimal" to "accurate". Our reasoning is following:
On the one hand, we use ADMM to optimize convex function with convex constraints (and thus, our algorithm should converge to optimum),
On the other hand, we use a fixed number of iterations, so convergence is not guaranteed. Also, while our weight update might be optimal, we also use a mask selection heuristic, which is definitely not optimal. Thus, to prevent any overselling, we will stick to the term "accurate."

We are also replying to other concerns in comments to individual reviewers.

Finally, we uploaded a revised version of the paper, with new text highlighted in blue, changed text highlighted in green, and deleted text highlighted in red.

---

### Decision · Action_Editor_5tE2 · 2024-07-16

**Recommendation:** Accept with minor revision

**Comment:**

The reviewers agree that the accuracy and efficiency claims are supported with strong empirical evidence. The authors already made some changes to the paper based on the feedback from the reviewers (e.g., better related work, changing some terms around optimality). Below I summarize some additional updates that are important to implement for the final version of the paper to improve clarity and readability.

As the sections 2.3 and 3 are written right now, it is difficult for the reader to separate precise mathematical statements and informal interpretations and discussion:

 - Section 2.3 quotes results that are not directly used: only the key results that you rely on from Boyd et al. should be quoted. For example, there is no point in introducing function g(x) since it just maps to zero. I would suggest having a proposition statement in a form that you use (you are welcome to include the full theorem in the appendix). It is also crucial to state the exact conditions under which the algorithm converges to the optimum.

 - The application of the result by Boyd et al to pruning, as in Section 3, should be stated as a precise corollary (with a proof). Then the discussion can follow.


As mentioned by other reviewers, it would be great in Section 3 to explicitly contrast your proposed method to SparseGPT and Wanda (explaining how similar it is to SparseGPT and which component is being replaced).

Regarding the optimality and accuracy statements:  from what I can tell, there is no “accuracy” guarantee (no theorem saying that within k updates, one will be epsilon far from the most “accurate” solution, where the latter implicitly implies connections to generalization). Thus I do not think that replacing “optimal” with “accurate” is any better. Going back to optimality, even ignoring convergence, the optimal solution might only be reached for the objective based on the subsample of the training data, and also only for the convex per-layer subproblem which is not the same as optimality of pruning and updating the entire network. So I do agree with the reviewers that saying that the proposed algorithm is optimal in any sense is not accurate.

Some additional minor comments: (1) the way that the abstract is currently written makes it sound like you are proposing a better fine-tuning technique only (that should be applied to an already-sparse model). (2) In the introduction, under “Our results”, you say “Our algorithm sidesteps all of the problems of previous solutions.”. Please list these problems explicitly, as they are entangled in the previous paragraph and hard to separate.

**Audience:**

Efficient pruning of LLMs while maintaining performance is an important problem in large scale deep learning. The results presented in the paper introducing optimization tricks to improve upon existing pruning methods will be of interest to TMLR audience.

**Claims And Evidence:**

The paper proposes a new pruning algorithm that acts layer-wise and scales to large networks (large language models). The pruning can be done in one-shot, or gradually. The authors claim that their method requires fewer updates, and is more computationally efficient compared to recently-proposed alternatives (SparseGPT, Wanda), and provide empirical support for these claims.

---

> ### Author Response · Authors · 2024-07-22
>
> Dear action editor,
> thank you for accepting our paper with minor revisions.
>
> We revised the content according to your recommendations.